# An Empirical Comparison of Off-policy Prediction Learning Algorithms on the Collision Task

## Abstract

Off-policy prediction—learning the value function for one policy from data generated while following another policy—is one of the most challenging subproblems in reinforcement learning. This paper presents empirical results with eleven prominent off-policy learning algorithms that use linear function approximation: five Gradient-TD methods, two Emphatic-TD methods, Off-policy TD($\lambda$), Vtrace, and variants of Tree Backup and ABQ that are derived in this paper such that they are applicable to the prediction setting. Our experiments used the Collision task, a small off-policy problem analogous to that of an autonomous car trying to predict whether it will collide with an obstacle. We assessed the performance of the algorithms according to their learning rate, asymptotic error level, and sensitivity to step-size and bootstrapping parameters. By these measures, the eleven algorithms can be partially ordered on the Collision task. In the top tier, the two Emphatic-TD algorithms learned the fastest, reached the lowest errors, and were robust to parameter settings. In the middle tier, the five Gradient-TD algorithms and Off-policy TD($\lambda$) were more sensitive to the bootstrapping parameter. The bottom tier comprised Vtrace, Tree Backup, and ABQ; these algorithms were no faster and had higher asymptotic error than the others. Our results are definitive for this task, though of course experiments with more tasks are needed before an overall assessment of the algorithms' merits can be made.

## 1 Introduction

In reinforcement learning, it is not uncommon to learn the value function for one policy while following another policy. For example, the Q-learning algorithm (Watkins, 1989; Watkins & Dayan, 1992) learns the value of the greedy policy while the agent may select its actions according to a different, more exploratory, policy. The first policy, the one whose value function is being learned, is called the *target policy* while the more exploratory policy generating the data is called the *behavior policy*. When these two policies are different, as they are in Q-learning, the problem is said to be one of *off-policy learning*, whereas if they are the same, the problem is said to be one of *on-policy learning*. The former is 'off' in the sense that the data is from a different source than the target policy, whereas the latter is from data that is 'on' the policy. Off-policy learning is more difficult than on-policy learning and subsumes it as a special case.

There are various reasons for interest in off-policy learning. One reasons is that it has been the core of many of the great successes that have come out of the Deep Reinforcement Learning field in the past few years. Probably one of the most notable examples is the DQN architecture, in which the Q-learning algorithm was used to learn how to play Atari games (Mnih et al., 2015).

Another reason for interest in off-policy learning is that it provides a clear way of intermixing exploration and exploitation. The dilemma is that an agent should always *exploit* what it has learned so far—it should take the best actions according to what it has learned—but it should also always *explore* to find actions that might be superior. No agent can simultaneously behave in both ways. However, an off-policy algorithm can, in a sense, pursue both goals at the same time. The behavior policy can explore freely while the target policy can converge to the fully exploitative, optimal policy independent of the behavior policy's explorations.

Another appealing aspect of off-policy learning is that it enables learning about many policies in parallel. Once the target policy is freed from behavior, there is no reason to have a single target policy. With off-policy learning, an agent could simultaneously learn how to optimally perform many different tasks (as suggested by Jaderberg et al. (2016) and Rafiee et al. 2019). Parallel off-policy learning of value functions has even been proposed as a way of learning general, policy-dependent, world knowledge (e.g., Sutton et al., 2011; White, 2015; Ring, in prep). Finally, note that numerous ideas in the machine learning rely on off-policy learning, including the learning of temporally-abstract world models (Sutton, Precup, & Singh, 1999), predictive representations of state (Littman, Sutton, & Singh, 2002; Tanner & Sutton, 2005), auxiliary tasks (Jaderberg et al., 2016), life-long learning (White, 2015), and learning from historical data (Thomas, 2015).

Many off-policy learning algorithms have been explored in the history of reinforcement learning. Q-learning (Watkins, 1989; Watkins & Dayan, 1992) is perhaps the oldest. In the 1990s it was realized that combining off-policy learning, function approximation, and temporal-difference (TD) learning risked instability (Baird, 1995). Precup, Sutton, and Singh (2000) introduced off-policy algorithms with importance sampling and eligibility traces, as well as tree backup algorithms, but did not provide a practical solution to the risk of instability. Gradient-TD methods (see Maei, 2011; Sutton et al., 2009) assured stability by following the gradient of an objective function, as suggested by Baird (1999). Emphatic-TD methods (Sutton, Mahmood, & White, 2016) reweighted updates in such a way as to regain the convergence assurances of the original on-policy TD algorithms. These methods had convergence guarantees, but provide no assurances for efficiency in practice. Other algorithms, including Retrace (Munos et al., 2016), Vtrace (Espeholt et al., 2018) and ABQ (Mahmood, Yu, & Sutton, 2017) were developed recently to overcome difficulties encountered in practice.

As more off-policy algorithms were developed, there was a need to compare them systematically. However, comparing algorithms fairly within a DQN-like architecture was not possible. In a DQN-like architecture, many elements work in concert to solve a task. Each element has one or more parameters that need tuning. On one hand, not all these parameters can be tuned systematically due to the computational cost, and on the other hand, tuning parameters carefully and studying performance over many parameters is necessary for a fair comparative study. In the original DQN work, for example, the parameters were not systematically tuned due to the computational burden; the DQN paper reads: "The values of all the hyperparameters and optimizer parameters were selected by performing an informal search on the games Pong, Breakout, Seaquest, Space Invaders and Beam Rider." (Mnih et al., 2015). Due to the computational cost, to be able to conduct a fair and detailed comparative study, separate parts of a DQN-like architecture need to be studied alone.

We reduce the amount of required computation in this study in three ways. First, we focus on comparing off-policy algorithms and remove other confounding factors from the comparison. This means that the comparison will not include elements such as complex optimizers, target networks, or experience replay buffers. Second, we focus on linearly learning the value function from given and fixed features. These learned value functions can later be used for control. Focusing on linearly learning the value function through fixed features is justified through the two time scale view of Neural Networks (NNs) as described by Chung et al. (2018). In this view, it is assumed that the features are learned using the first $n-1$ layers of the neural network at their own time scale, and then the features are used by the last layer to linearly learn the value function. Third, we focus on fully incremental online algorithms. Many algorithms referred to as the *OPE* family of algorithms assume access to data beyond what the agent experiences at each time step. Our paper, focuses on the fully incremental setting, in which the agent makes one interaction with the environment, receives a reward, learns from it, and then discards the sample and moves to the next step. This is in contrast to the setting in which the agent has access to historical data. Not having access to historical data, the agent is more limited in what it can learn.

In fact, there have been a few empirical studies that compare off-policy prediction learning algorithms in small environments. The earliest systematic study was that by Geist and Scherrer (2014). Their experiments were on random MDPs and compared eight off-policy algorithms. A few months later, Dann, Neumann, and Peters (2014) published a more in-depth study with one additional algorithm and six test problems including random MDPs. Both studies considered off-policy problems in which the target and behavior policies were given and stationary. Such *prediction* problems allow for relatively simple experiments and are still challenging (e.g., they involve the same risk of instability). Both studies used linear function approximation with a given feature representation. The algorithms studied by Geist and Scherrer (2014), and by Dann, Neumann, and

Peters (2014) can be divided into those whose per-step complexity is linear in the number of parameters, like TD($\lambda$), and methods whose complexity is quadratic in the number of parameters (proportional to the square of the number of parameters), like Least Squares TD($\lambda$) (Bradtke & Barto, 1996; Boyan, 1999). Quadratic-complexity methods avoid the risk of instability, but cannot be used in learning systems with large numbers (e.g., millions) of weights. A third systematic study, by White and White (2016), excluded quadratic-complexity algorithms, but added four additional linear-complexity algorithms.

The current paper is similar to previous studies in that it treats prediction with linear function approximation, and similar to the study by White and White (2016) in restricting attention to linear complexity algorithms. Our study differs from earlier studies in that it treats more algorithms and does a deeper empirical analysis on a single problem, the Collision task. The additional algorithms are the prediction variants of Tree Backup($\lambda$) (Precup, Sutton, & Singh, 2000), Retrace($\lambda$) (Munos et al., 2016), ABQ($\zeta$) (Mahmood, Yu, & Sutton, 2017), and TDRC($\lambda$) (Ghiassian et al., 2020). Our empirical analysis is deeper primarily in that we examine and report the dependency of all eleven algorithms' performance on all of their parameters individually. This level of detail is needed to expose our main result, an overall ordering of the performance of off-policy algorithms on the Collision task. Our results, though limited to this task, are a significant addition to what is known about the comparative performance of off-policy learning algorithms.

## 2 Formal Framework

In this section, we formally explain the framework of off-policy prediction learning with linear function approximation. An agent and environment interact at discrete time steps, $t = 0, 1, 2, \ldots$. The environment is a Markov Decision Process (MDP) with state $S_t \in \mathcal{S}$ at time step $t$. At each time step, the agent chooses an action $A_t \in \mathcal{A}$ with probability $b(a|s)$, where the function $b \colon \mathcal{A} \times \mathcal{S} \to [0, 1]$ with $\sum_{a \in \mathcal{A}} b(a|s) = 1, \forall s \in \mathcal{S}$, is called the *behavior* policy because it determines the agent's behavior. After taking action $A_t$ in state $S_t$, the agent receives from the environment a numerical reward $R_{t+1} \in \mathcal{R} \subset \mathbb{R}$ and the next state $S_{t+1}$. In general the reward and next state are stochastically jointly determined by the current state and action.

In prediction learning, we estimate for each state the expected discounted sum of future rewards, given that actions are taken according to a different policy $\pi$, called the *target* policy (because learning its values is the target of our learning). For simplicity, both target and behavior policies are assumed here to be known and static, although of course in many applications of interest one or the other may be changing. The discounted sum of future rewards at time $t$ is called the *return* and denoted $G_t$:

$$G_t \overset{\mathrm{def}}{=} R_{t+1} + \gamma R_{t+2} + \gamma^2 R_{t+3} + \cdots$$

The expected return when starting from a state and following a specific policy thereafter is called the *value* of the state under the policy. The *value function* $v_\pi : \mathcal{S} \to \mathbb{R}$ for a policy $\pi$ takes a state as input and returns the value of that state:

$$v_\pi(s) \overset{\mathrm{def}}{=} \mathbb{E}[G_t \mid S_t = s, A_{t:\infty} \sim \pi]. \tag{1}$$

Prediction learning algorithms seek to learn an estimate $\hat{v} : \mathcal{S} \to \mathbb{R}$ that approximates the true value function $v_\pi$. In many problems $\mathcal{S}$ is large and an exact approximation is not possible even in the limit of infinite time and data. Many parametric forms are possible, including deep artificial neural networks, but of particular interest, and our exclusive focus here, is the linear form:

$$\hat{v}(s, \mathbf{w}) \overset{\mathrm{def}}{=} \mathbf{w}^\top \mathbf{x}(s), \tag{2}$$

where $\mathbf{w} \in \mathbb{R}^d$ is a learned weight vector and $\mathbf{x}(s) \in \mathbb{R}^d, \forall s \in \mathcal{S}$ is a set of given feature vectors, one per state, where $d \ll |\mathcal{S}|$.

## 3 Algorithms

In this section, we briefly introduce the eleven algorithms used in our empirical study. These eleven are intended to include all the best candidate algorithms for off-policy prediction learning with linear function

approximation. The complete update rules of all algorithms and additional technical discussion can be found in Appendix A and Appendix E respectively. Many algorithms studied in our paper can be combined with each other; for example, the combination of Emphatic TD and Gradient TD results in Emphatic Gradient TD. Similarly the combination of Gradient TD and Tree Backup results in GTB (Touati, 2018). We did not include algorithm combinations to keep the scope focused.

*Off-policy TD(λ)* (Precup, Sutton, & Dasgupta, 2001) is the off-policy variant of the original TD(λ) algorithm (Sutton, 1988) that uses importance sampling to reweight the returns and account for the differences between the behavior and target policies. This algorithm has just one set of weights and one step-size parameter.

Our study includes five algorithms from the Gradient-TD family. *GTD(λ)* and *GTD2(λ)* are based on algorithmic ideas introduced by Sutton et al., (2009), then extended to eligibility traces by Maei (2011). *Proximal GTD2(λ)* (Mahadevan et al., 2014; Liu et al., 2015; Liu et al., 2016) is a "mirror descent" version of GTD2 using a saddle-point objective function. These algorithms approximate stochastic gradient descent (SGD) on an alternative objective function, the mean squared projected Bellman error. *HTD(λ)* (Hackman, 2012; White & White, 2016) is a "hybrid" of GTD(λ) and TD(λ) which becomes equivalent to classic TD(λ) where the behavior policy coincides with the target policy. *TDRC(λ)* is a recent variant of GTD(λ) that adds regularization. All these methods involve an additional set of learned weights (beyond that used in $\hat{v}$) and a second step-size parameter, which can complicate their use in practice. TDRC(λ) offers a standard way of setting the second step-size parameter, which makes this less of an issue. All of these methods are guaranteed to converge with an appropriate setting of their two step-size parameters.

Our study includes two algorithms from the Emphatic-TD family. Emphatic-TD algorithms attain stability by up- or down-weighting the updates made on each time step by Off-policy TD(λ). If this variation in the emphasis of updates is done in just the right way, stability can be guaranteed with a single set of weights and a single step-size parameter. The original emphatic algorithm, *Emphatic TD(λ)*, was introduced by Sutton, Mahmood, and White (2016). The variant *Emphatic TD(λ, β)*, introduced by Hallak et al., (2016), has an additional parameter, $\beta \in [0, 1]$, intended to reduce variance.

The final three algorithms in our study—ABTD(ζ), Vtrace(λ), and the prediction variant of Tree Backup(λ)—can be viewed as attempts to address the problem of large variations in the product of importance sampling ratios. If this product might become large, then the step-size parameter must be set small to ensure there is no overshoot—and then learning may be slow. All these methods attempt to control the importance sampling product by changing the bootstrapping parameter from step to step (Yu, Mahmood, & Sutton, 2018). Munos et al., (2016) proposed simply putting a cap on the importance sampling ratio at each time step; they explored the theory and practical consequences of this modification in a control context with their Retrace algorithm. *Vtrace(λ)* (Espeholt et al., 2018) is a modification of Retrace to make it suitable for prediction rather than control. Mahmood, Yu, and Sutton (2017) developed a more flexible algorithm that achieves a similar effect. Their algorithm was also developed for control; to apply the idea to prediction learning we had to develop a nominally new algorithm, *ABTD(ζ)*, that naturally extends ABQ(ζ) from control to prediction. ABTD(ζ) will be developed in the next section. Finally, *Tree Backup(λ)* (Precup, Sutton, & Singh, 2000) reduces the effective λ by the probability of the action taken at each time step. Each of these algorithms (or their control predecessors) have been shown to be very effective on specific problems.

## 4 Derivations of Tree Backup, Vtrace, and ABTD

In this section, we derive the prediction variants of Tree Backup(λ), Retrace(λ), and ABQ(ζ). The prediction variant of Tree Backup(λ) and the prediction variant of ABQ(ζ), which we call ABTD(ζ), are new to this paper. Vtrace(λ) is not a new algorithm, and was previously discussed by Espeholt et al., (2018). In this paper, we use the procedure suggested by Mahmood, Yu, and Sutton (2017) to arrive, in a new way, at the same Vtrace algorithm derived by Espeholt et al., (2018). We will additionally show that all three algorithms can be seen as Off-policy TD(λ) with $\lambda_t$ generalized from a constant to a function of $(S_t, A_t)$. Readers who are only interested in the relative performance of algorithms and practical issues, can skip this section.

Deriving the prediction variant of control algorithms is typically straightforward. However, deriving the prediction variant of the three mentioned algorithms is a little more involved. The three control algorithms—

ABQ($\zeta$), Retrace($\lambda$), and the control variant of Tree Backup($\lambda$)—avoid all importance sampling ratios in their update rules to stabilize learning. As we will shortly see, importance sampling ratios cannot be completely avoided in the prediction setting as was done in the control setting. Trying to avoid all importance sampling ratios in the prediction learning case might result in an incorrect version of these algorithms that we will discuss in Section 4.1.

The prediction variant of all three algorithms can be derived in a similar way. To understand the prediction variant of these algorithms, we derive ABTD($\zeta$). We then use ABTD($\zeta$) to derive extensions to Vtrace($\lambda$) and Tree Backup($\lambda$) for prediction. The key idea is to set $\lambda_t = \lambda(S_{t-1}, A_{t-1})$ adaptively in generic Off-policy TD($\lambda$):

$$\mathbf{z}_t \leftarrow \rho_{t-1}\gamma_t\lambda_t\mathbf{z}_{t-1} + \mathbf{x}_t \quad \text{with } \mathbf{z}_{-1} = \mathbf{0} \tag{3}$$

$$\mathbf{w}_{t+1} \leftarrow \mathbf{w}_t + \alpha\rho_t\delta_t\mathbf{z}_t, \tag{4}$$

where $\delta_t$ is the TD-error, $\mathbf{z}_t$ is eligibility trace, $\rho_t \overset{\text{def}}{=} \pi(A_t|S_t)/b(A_t|S_t)$ is the importance sampling ratio, and $\alpha$ is the step-size parameter. The update rules provided here for Off-policy TD($\lambda$), are different from the following update rules often provided in the literature for Off-policy TD($\lambda$):

$$\mathbf{z}_t \leftarrow \rho_t(\gamma_t\lambda_t\mathbf{z}_{t-1} + \mathbf{x}_t) \quad \text{with } \mathbf{z}_{-1} = \mathbf{0} \tag{5}$$

$$\mathbf{w}_{t+1} \leftarrow \mathbf{w}_t + \alpha\delta_t\mathbf{z}_t. \tag{6}$$

In Appendix B we show that these two sets of update rules are the same numerically step by step. We use the first set of update rules here as they are more appropriate for our purposes in this paper.

Consider the generalized $\lambda$-return, for a $\lambda$ based on the state and action—as in ABQ($\zeta$)—or the entire transition (White, 2017). Let $\lambda_{t+1} = \lambda(S_t, A_t, S_{t+1})$ be defined based on the transition $(S_t, A_t, S_{t+1})$, corresponding to how rewards and discounts are defined based on the transition, $R_{t+1} = r(S_t, A_t, S_{t+1})$ and $\gamma_{t+1} = \gamma(S_t, A_t, S_{t+1})$. Then, given a value function $\hat{v}$, the $\lambda$-return $G_t^\lambda$ for generalized $\gamma$ and $\lambda$ is defined recursively as

$$G_t^\lambda \overset{\text{def}}{=} \rho_t\left(R_{t+1} + \gamma_{t+1}\left[(1-\lambda_{t+1})\hat{v}(S_{t+1}) + \lambda_{t+1}G_{t+1}^\lambda\right]\right).$$

Similar to ABQ($\zeta$) (Mahmood et al., 2017, Equation 7), this $\lambda$-return can be written using TD-errors

$$\delta_t \overset{\text{def}}{=} R_{t+1} + \gamma_{t+1}\hat{v}(S_{t+1}) - \hat{v}(S_t),$$

as

$$\begin{aligned}
G_t^\lambda &= \rho_t\left(R_{t+1} + \gamma_{t+1}\hat{v}(S_{t+1}) - \gamma_{t+1}\lambda_{t+1}\hat{v}(S_{t+1}) + \gamma_{t+1}\lambda_{t+1}G_{t+1}^\lambda\right) \\
&= \rho_t\left(\delta_t + \hat{v}(S_t) + \gamma_{t+1}\lambda_{t+1}\left[G_{t+1}^\lambda - \hat{v}(S_{t+1})\right]\right) \\
&= \rho_t\delta_t + \rho_t\hat{v}(S_t) + \rho_t\gamma_{t+1}\lambda_{t+1}\left(\rho_{t+1}\delta_{t+1} + \rho_{t+1}\gamma_{t+2}\lambda_{t+2}\left[G_{t+2}^\lambda - \hat{v}(S_{t+2})\right]\right) \\
&= \rho_t\sum_{n=t}^{\infty}(\rho_{t+1}\lambda_{t+1}\gamma_{t+1})^n\delta_t + \rho_t\hat{v}(S_t),
\end{aligned}$$

where we define $(\rho_{t+1}\lambda_{t+1}\gamma_{t+1})^n \overset{\text{def}}{=} \prod_{i=t+1}^{n}\rho_i\lambda_i\gamma_i$.

This return differs from the return used by ABQ($\zeta$), because it corresponds to the return from a state, rather than the return from a state and action. In ABQ($\zeta$), the goal is to estimate the action-value for a given state and action. For ABTD($\zeta$), the goal is to estimate the value for a given state. For the return from a state $S_t$, we need to correct the distribution over actions $A_t$ with importance sampling ratio $\rho_t$. For ABQ($\zeta$), the correction with $\rho_t$ is not necessary because $S_t$ and $A_t$ are both given, and importance sampling corrections only need to be computed for future states and actions, with $\rho_{t+1}$ onward. For ABTD($\zeta$), therefore, unlike ABQ($\zeta$), not all importance sampling ratios can be avoided. We can, however, still set $\lambda$ in a similar way to ABQ($\zeta$) to mitigate the variance effects of importance sampling.

To ensure $\rho_t\lambda_{t+1}$ is well-behaved, ABTD($\zeta$) sets $\lambda$ as follows:

$$\lambda(S_t, A_t, S_{t+1}) = \nu(\psi, S_t, A_t)b(S_t, A_t),$$

with the following scalar parameters to define $\nu_t$ (Mahmood, Yu, & Sutton, 2017):

$$\nu_t \stackrel{\text{def}}{=} \nu(\psi(\zeta), S_t, A_t) \stackrel{\text{def}}{=} \min\left(\psi(\zeta), \frac{1}{\max(b(A_t|S_t), \pi(A_t|S_t))}\right),$$

$$\psi(\zeta) \stackrel{\text{def}}{=} 2\zeta\psi_0 + \max(0, 2\zeta - 1)(\psi_{\max} - 2\psi_0),$$

$$\psi_0 \stackrel{\text{def}}{=} \frac{1}{\max_{s,a} \max(b(a|s), \pi(a|s))},$$

$$\psi_{\max} \stackrel{\text{def}}{=} \frac{1}{\min_{s,a} \max(b(a|s), \pi(a|s))}.$$

In the $\lambda$-return, then

$$\rho_t\lambda_{t+1} = \frac{\pi(S_t, A_t)}{b(S_t, A_t)}\nu(\psi, S_t, A_t)b(S_t, A_t) = \nu(\psi, S_t, A_t)\pi(S_t, A_t).$$

This removes the importance sampling ratios from the eligibility trace. The resulting ABTD($\zeta$) algorithm can be written as the standard Off-policy TD($\lambda$) algorithm, for a particular setting of $\lambda$. The Off-policy TD($\lambda$) algorithm, with this $\lambda$, is called ABTD($\zeta$), with updates

$$\delta_t \stackrel{\text{def}}{=} R_{t+1} + \gamma_{t+1}\mathbf{w}_t^\top\mathbf{x}_{t+1} - \mathbf{w}_t^\top\mathbf{x}_t$$

$$\mathbf{z}_t \leftarrow \gamma_t\nu_{t-1}\pi_{t-1}\mathbf{z}_{t-1} + \mathbf{x}_t \quad \text{with } \mathbf{z}_{-1} = \mathbf{0}$$

$$\mathbf{w}_{t+1} \leftarrow \mathbf{w}_t + \alpha\rho_t\delta_t\mathbf{z}_t.$$

Finally, we can adapt Retrace($\lambda$) and Tree Backup($\lambda$) for policy evaluation. Mahmood, Yu, and Sutton (2017) showed that Retrace($\lambda$) can be specified with a particular setting of $\nu_t$ (in their Equation 36). We can similarly obtain Retrace($\lambda$) for prediction by setting

$$\nu_{t-1} = \zeta \min\left(\frac{1}{\pi_{t-1}}, \frac{1}{b_{t-1}}\right),$$

or more generally:

$$\nu_{t-1} = \zeta \min\left(\frac{\bar{c}}{\pi_{t-1}}, \frac{1}{b_{t-1}}\right),$$

where $\bar{c}$ is a constant, which we will discuss in more detail shortly. For Tree Backup($\lambda$), the setting for $\nu_t$ is any constant value in $[0, 1]$ (see Algorithm 2 of Precup, Sutton & Singh, 2000).

So far, we derived ABTD($\zeta$) for prediction by defining $\lambda_t$ in the eligibility trace update of Off-policy TD($\lambda$). We then used two special settings of $\nu$ to recover Vtrace($\lambda$) and Tree Backup($\lambda$) algorithms. Now, we specify Tree Backup($\lambda$), and Vtrace($\lambda$) updates again, but this time in terms of a special setting of $\lambda_t$ in the Off-policy TD($\lambda$) update.

Prediction variant of Tree Backup($\lambda$) is Off-policy TD($\lambda$) with $\lambda_t = b_{t-1}\lambda$, for some tuneable constant $\lambda \in [0, 1]$. Replacing $\lambda_t$ with $b_{t-1}\lambda$ in the eligibility trace update in (3) simplifies as follows:

$$\mathbf{z}_t \leftarrow \gamma_t\frac{\pi_{t-1}}{b_{t-1}}b_{t-1}\lambda\mathbf{z}_{t-1} + \mathbf{x}_t,$$

$$= \gamma_t\pi_{t-1}\lambda\mathbf{z}_{t-1} + \mathbf{x}_t. \tag{7}$$

A simplified variant of the Vtrace($\lambda$) algorithm (Espeholt et al., 2018) can be derived with a similar substitution:

$$\lambda_t = \min\left(\frac{\bar{c}}{\pi_{t-1}}, \frac{1}{b_{t-1}}\right)\lambda b_{t-1},$$

where $\bar{c} \in \mathbb{R}^+$ and $\lambda \in [0, 1]$ are both tuneable constants. The update rule for the eligibility trace of Vtrace($\lambda$) with this special setting of $\lambda_t$ at each time step becomes:

$$
\begin{aligned}
\mathbf{z}_t &\leftarrow \gamma_t \min\left(\frac{\bar{c}}{\pi_{t-1}}, \frac{1}{b_{t-1}}\right) \lambda b_{t-1} \frac{\pi_{t-1}}{b_{t-1}} \mathbf{z}_{t-1} + \mathbf{x}_t \\
&= \gamma_t \min\left(\frac{\bar{c}}{\pi_{t-1}}, \frac{1}{b_{t-1}}\right) \lambda \pi_{t-1} \mathbf{z}_{t-1} + \mathbf{x}_t \\
&= \gamma_t \min\left(\frac{\bar{c}\pi_{t-1}}{\pi_{t-1}}, \frac{\pi_{t-1}}{b_{t-1}}\right) \lambda \mathbf{z}_{t-1} + \mathbf{x}_t \\
&= \gamma_t \min\left(\bar{c}, \rho_{t-1}\right) \lambda \mathbf{z}_{t-1} + \mathbf{x}_t.
\end{aligned}
\tag{8}
$$

The parameter $\bar{c}$ is used to clip importance sampling ratios in the trace. Note that it is not possible to recover the full Vtrace($\lambda$) algorithm in this way. The more general Vtrace($\lambda$) algorithm uses an additional parameter, $\bar{\rho} \in \mathbb{R}^+$ that clips the $\rho_t$ in the update to $\mathbf{w}_{t+1}$: $\min(\bar{\rho}, \rho_t)\delta_t \mathbf{z}_t$. When $\bar{\rho}$ is set to the largest possible importance sampling ratio, it does not affect $\rho_t$ in the update to $\mathbf{w}_t$ and so we obtain the equivalence above. For smaller $\bar{\rho}$, however, Vtrace($\lambda$) is no longer simply an instance of Off-policy TD($\lambda$). In our experiments, we investigate this simplified variant of Vtrace($\lambda$) that does not clip $\rho_t$ and set $\bar{c} = 1$ as done in the original Retrace algorithm.

Finally, as mentioned before, ABTD($\zeta$) for $\zeta \in [0, 1]$ uses $\lambda_t = \nu_{t-1}b_{t-1}$ in the Off-policy TD($\lambda$) update which results in the following eligibility trace update:

$$
\begin{aligned}
\mathbf{z}_t &\leftarrow \gamma_t \frac{\pi_{t-1}}{b_{t-1}} \nu_{t-1} b_{t-1} \mathbf{z}_{t-1} + \mathbf{x}_t \\
&= \gamma_t \nu_{t-1} \pi_{t-1} \mathbf{z}_{t-1} + \mathbf{x}_t,
\end{aligned}
\tag{9}
$$

The convergence properties of all three methods are similar to Off-policy TD($\lambda$). They are not guaranteed to converge under off-policy sampling with weighting $\mu_b$ and function approximation. With the addition of gradient corrections similar to GTD($\lambda$), all three algorithms are convergent. For explicit theoretical results, see Mahmood, Yu, and Sutton (2017) for ABQ($\zeta$) with gradient correction and Touati et al. (2018) for convergent versions of Retrace($\lambda$) and Tree Backup($\lambda$).

## 4.1 An alternative but incorrect extension of ABQ($\zeta$) to ABTD($\zeta$)

The ABQ($\zeta$) algorithm specifies $\lambda$ to ensure that $\rho_t \lambda_t$ is well-behaved, whereas we specified $\lambda$ so that $\rho_t \lambda_{t+1}$ is well-behaved. This difference arises from the fact that for action-values, the immediate reward and next state are not re-weighted with $\rho_t$. Consequently, the $\lambda$-return of a policy from a given state and action is:

$$
R_{t+1} + \gamma_{t+1}\left[(1 - \lambda_{t+1})\hat{v}(S_{t+1}) + \rho_{t+1}\lambda_{t+1}G_{t+1}^\lambda\right].
$$

To mitigate variance in ABQ($\zeta$) when learning action-values, therefore, $\lambda_{t+1}$ should be set to ensure that $\rho_{t+1}\lambda_{t+1}$ is well-behaved. For ABTD($\zeta$), however, $\lambda_{t+1}$ should be set to mitigate variance from $\rho_t$ rather than from $\rho_{t+1}$.

To see why more explicitly, the central idea of these algorithms is to avoid importance sampling altogether: this choice ensures that the eligibility trace does not include importance sampling ratios. The eligibility trace $\mathbf{z}_t^a$ in TD when learning action-values is:

$$
\mathbf{z}_t^a = \rho_t \lambda_t \gamma_t \mathbf{z}_{t-1}^a + \mathbf{x}_t^a,
$$

for state-action features $\mathbf{x}_t^a$. For $\rho_t \lambda_t = \nu_t \pi_t$, this trace reduces to $\mathbf{z}_t^a = \nu_t \pi_t \gamma_t \mathbf{z}_{t-1}^a + \mathbf{x}_t^a$ (Equation 18, Mahmood et al., 2017). For ABTD($\zeta$), one could in fact also choose to set $\lambda_t$ so that $\rho_t \lambda_t = \nu_t \pi_t$ instead of $\rho_t \lambda_{t+1} = \nu_t \pi_t$. However, this would result in eligibility traces that still contain importance sampling ratios. The eligibility trace in TD when learning state-values is:

$$
\mathbf{z}_t = \rho_{t-1} \lambda_t \gamma_t \mathbf{z}_{t-1} + \mathbf{x}_t.
$$

Setting $\rho_t \lambda_t = \nu_t \pi_t$ would result in the update $\mathbf{z}_t = \rho_{t-1}\nu_t \frac{\pi_t}{\rho_t}\gamma_t \mathbf{z}_{t-1} + \mathbf{x}_t$, which does not remove important sampling ratios from the eligibility trace.

## 5 Collision Task

The Collision task is an idealized off-policy prediction-learning task. A vehicle moves along an eight-state track towards an obstacle with which it will collide if it keeps moving forward. In this episodic task, each episode begins with the vehicle in one of the first four states (selected at random with equal probability). In these four states, `forward` is the only possible action whereas, in the last four states, two actions are possible: `forward` and `turnaway` (see Figure 1). The `forward` action always moves the vehicle one state further along the track; if it is taken in the last state, then a collision is said to occur, the reward is 1, and the episode ends. The `turnaway` action causes the vehicle to "turn away" from the wall, which also ends the episode, except with a reward of zero. The reward is also zero on all earlier, non-terminating transitions. In an episodic task like this the return is accumulated only up to the end of the episode. After termination, the next state is the first state of the next episode, selected randomly from the first four as specified above.

The target policy on this task is to always take the `forward` action, $\pi(\texttt{forward}|s) = 1, \forall s \in \mathcal{S}$, whereas the behavior policy is to take the two actions (where available) with equal probability, $b(\texttt{forward}|s) = b(\texttt{turnaway}|s) = 0.5, \forall s \in \{5, 6, 7, 8\}$. The problem is discounted with a discount rate of $\gamma = 0.9$. As always, we are seeking to learn the value function for the target policy, which in this case is $v_\pi(s) = \gamma^{8-s}$. This function is shown as a dashed black line in Figure 2. The thin red lines show approximate value functions $\hat{v} \approx v_\pi$, using various feature representations, as we discuss shortly below.

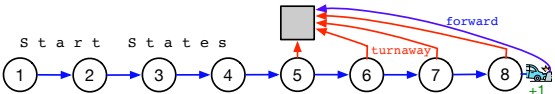

Figure 1: The Collision task. Episodes start in one of the first four states and end when the `forward` action is taken from the eighth state, causing a crash and a reward of 1, or when the `turnaway` action is taken in one of the last four states.

This idealized task is roughly analogous to and involves some similar issues as real-world autonomous driving problems, such as exiting a parallel parking spot without hitting the car in front of you, or learning how close you can get to other cars without risking collisions. In particular, if these problems can be treated as off-policy learning problems, then solutions can potentially be learned with fewer collisions. In this paper, we are testing the efficiency of various off-policy prediction-learning algorithms at maximizing how much they learn from the same number of collisions.

Similar problems have been studied using mobile robots. For example, White (2015) used off-policy learning algorithms running on an iRobot Create to predict collisions as signaled by activation of the robot's front bumper sensor. Rafiee et al. (2019) used a Kobuki robot to not only anticipate collisions, but to turn away from anticipated collisions before they occurred. Modayil and Sutton (2014) trained a custom robot to predict motor stalls and turn off the motor when a stall was predicted.

Our task is a prediction, and not a control task. If the task was a control task, the car would learn to hit the obstacle more often given our reward function. However, in our setting, the behavior and target policies are fixed and given, and the goal is to only learn about collisions. These predictions about collisions can later be used for different purposes such as state construction, and control. Through off-policy learning, the agent will be able to experience collisions, and predict with great detail, how close the agent is to the end of the corridor, without having to experience collisions many times. Without off-policy learning, the agent would have to experience collisions many more times in order to have accurate predictions about them.

We artificially introduce function approximation into the Collision task. Although a tabular approach is entirely feasible on this small problem, it would not be on the large problems of interest. In real applications, the agent would have sensor readings, which will go through an artificial neural network to create feature representations. We simulate such representations in the Collision task by randomly assigning to each of the eight states a binary feature vector $\mathbf{x}(s) \in \{0, 1\}^d, \forall s \in \{1..8\}$. We chose $d = 6$, so that was not possible for all eight of the feature vectors (one per state) to be linearly independent. In particular, we chose all eight feature vectors to have exactly three 1s and three 0s, with the location of the 1s for each state being chosen randomly.

Because the feature vectors are linearly dependent, it is not possible in general for a linear approximation, $\hat{v}(s, \mathbf{w}) = \mathbf{w}^\top \mathbf{x}$, to equal to $v_\pi(s)$ at all eight states of the Collision task. This, in fact, is the sole reason the red approximate value functions in Figure 2 do not exactly match $v_\pi$. Given a feature representation $\mathbf{x} : \mathcal{S} \to \mathbb{R}^d$, a linear approximate value function is completely determined by its weight vector $\mathbf{w} \in \mathbb{R}^d$. The quality of that approximation is assessed by its squared error at each state, weighted by how often each state occurs:

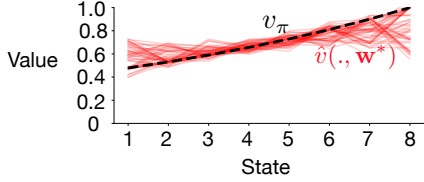

$$\overline{\text{VE}}(\mathbf{w}) = \sum_{s \in \mathcal{S}} \mu_b(s) \big[\hat{v}(s, \mathbf{w}) - v_\pi(s)\big]^2, \tag{10}$$

Figure 2: The ideal value function, $v_\pi$, and the best approximate value functions, $\hat{v}$, for 50 different feature representations.

where $\mu_b(s)$ is the state distribution, the fraction of time steps in which $S_t = s$, under the behavior policy (here $\mu_b$ was approximated from visitation counts from one million sample time steps). The value functions shown by red lines in Figure 2 are for $\mathbf{w}^*$, the weight vector that minimizes $\overline{\text{VE}}(\mathbf{w})$, with each line corresponding to a different randomly selected feature representation as described earlier. For these value functions, $\overline{\text{VE}}(\mathbf{w}^*) \approx 0.05$. All the code for the Collision task and the experiments are provided. See Appendix D.3.

## 6 Experiment

The Collision task, in conjunction with its behavior policy, was used to generate 20,000 time steps, comprising one *run*, and then this was repeated for a total of 50 independent runs. Each run also used a different feature representation randomly generated as described in the previous section. Focusing on one-hot representations, we decided to choose a different random representation for each of the 50 runs, to study the performance of algorithms across various one-hot representations. The eleven learning algorithms were then applied to the 50 runs, each with a range of parameter values; each combination of algorithm and parameter settings is termed an *algorithm instance*. A list of all parameter settings used can be found in Appendix C. They included 12 values of $\lambda$, 19 values of $\alpha$, 15 values of $\eta$ (for the Gradient-TD family), six values of $\beta$ (for ETD($\lambda, \beta$)), and 19 values of $\zeta$ (for ABTD($\zeta$)), for approximately 20,000 algorithm instances in total. In each run, the weight vector was initialized to $\mathbf{w}_0 = \mathbf{0}$ and then updated at each step by the algorithm instance to produce a sequence of $\mathbf{w}_t$. At each step we also computed and recorded $\overline{\text{VE}}(\mathbf{w}_t)$. In Deep Learning, it is important that the neural network is initialized using random weights because if not, the derivatives in backpropagation will be the same for all weights, and all learned features will be the same. In linear function approximation with given features this is not an issue, so we decided to initialize all weights to zero.

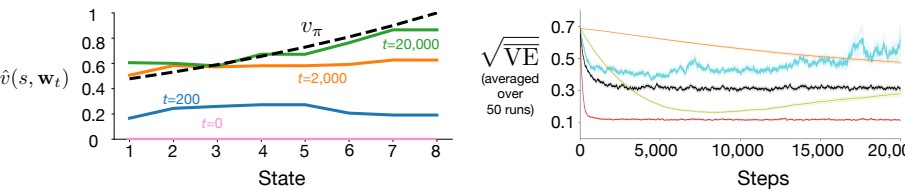

Figure 3: **Left:** An example of the approximate value function, $\hat{v}$, being learned over time. **Right:** Learning curves illustrating the range of things that can happen during a run. The average error over the 20,000 steps is a good combined measure of learning rate and asymptotic error.

With a successful learning procedure, we expect the value function to evolve over time as shown in the left panel of Figure 3. The approximate value function starts at $\hat{v}(s, \mathbf{0}) = 0$, as shown by the pink line, then moves toward positive values, as shown by the blue and orange lines. Finally, the learned value function slants and comes to closely approximate the true value function, though always with some residual error due to the limited feature representation, as shown by the green line (and also by all the red lines in Figure 2).

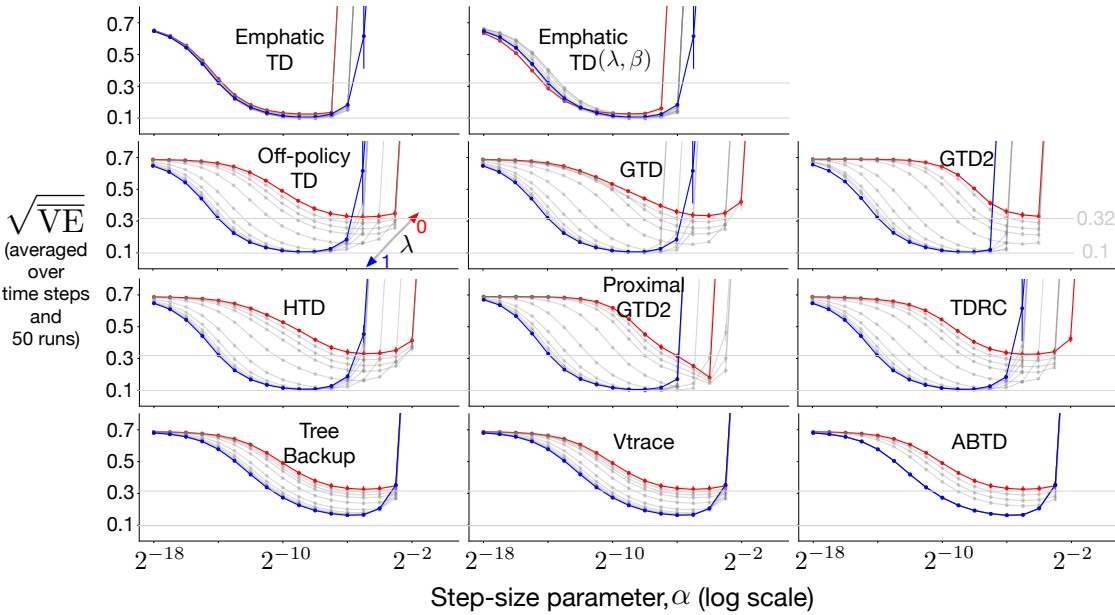

Figure 4: Performance of all algorithms on the Collision task as a function of their parameters $\alpha$ and $\lambda$. Each point is the average error over 50 runs. The bars over each point show the standard error. The red curves show the performance with $\lambda = 0$; the blue curves show the performance with $\lambda = 1$; and the gray curves show the performance with intermediate values of $\lambda$. The top tier algorithms (top row) attained a low error ($\approx 0.1$) at all $\lambda$ values. The middle tier of six algorithms attained a low error for $\lambda = 1$, but not for $\lambda = 0$. And the bottom-tier of three algorithms were unable to reach an error of $\approx 0.1$ at any $\lambda$ value.

The right panel of Figure 3 shows learning curves illustrating the range of things that happened in the experiment. Normally, we expect $\overline{\text{VE}}$ to decrease over the course of the experiment, starting at $\overline{\text{VE}}(\mathbf{0}) \approx 0.7$ and falling to some minimum value, as in the red and black lines in Figure 3 (these and all other data are averaged over the 50 runs). If the primary step-size parameter, $\alpha$, is small, then learning may be slow and incomplete by the end of the runs, as in the orange line. A larger step-size parameter may be faster, but, if it is too large, then divergence can occur, as in the blue line. For one algorithm, Proximal GTD2($\lambda$), we found that the error dipped low and then leveled off at a higher level, as in the olive line.

## 7 Main Results: A Partial Order over Algorithms

As an overall measure of the performance of an algorithm instance, we take its learning curve over 50 runs, as in Figure 3, and then average it across the 20,000 steps. In this way, we reduce all the data for an algorithm instance to a single number that summarizes performance. These numbers appear as points in our main results figure, Figure 4. Each panel of the figure is devoted to a single algorithm.

For example, performance numbers for instances of Off-policy TD($\lambda$) are shown as points in the left panel of the second row of Figure 4. This algorithm has two parameters, the step-size parameter, $\alpha$, and the bootstrapping parameter, $\lambda$. The points are plotted as a function of $\alpha$, and points with the same $\lambda$ value are connected by lines. The blue line shows the performances of the instances of Off-policy TD($\lambda$) with $\lambda = 1$, the red line shows the performances with $\lambda = 0$, and the gray lines show the performances with intermediate $\lambda$s. Note that all the lines are U-shaped functions of $\alpha$, as is to be expected; at small $\alpha$ learning is too slow to make much progress, and at large $\alpha$ there is overshoot and divergence, as in the blue line in Figure 3. For each point, the standard error over the 50 runs is also given as an error bar, though these are too small to be seen in all except the rightmost points of each line where the step size was highest and divergence was common. Except for these rightmost points, almost all visible differences are statistically significant.

First focus on the blue line (of the left panel on the second row of Figure 4), representing the performances of Off-policy TD($\lambda$) with $\lambda = 1$. There is a wide sweet spot, that is, there are many intermediate values of $\alpha$ at which good performance (low average error) is achieved. Note that the step-size parameter $\alpha$ is varied over a wide range, with logarithmic steps. The minimal error level of about 0.1 was achieved over four or five powers of two for $\alpha$. This is the primary measure of good performance that we look for in these data: low error over a wide range of parameter values.

Now contrast the blue line with the red and gray lines (for Off-policy TD($\lambda$) in the left panel of the second row of Figure 4). Recall that the blue line is for $\lambda = 1$, the red line is for $\lambda = 0$, and the gray lines are for intermediate values of $\lambda$. First note that the red line shows generally worse performance; the error level at $\lambda = 0$ was higher, and its range of good $\alpha$ values was slightly smaller (on a logarithmic scale). The intermediate values of $\lambda$ all had performances that were between the two extremes. Second, the sweet spot (the best $\alpha$ value) consistently shifted right, toward higher $\alpha$, as $\lambda$ was decreased from 1 toward 0.

Now, armed with a thorough understanding of the Off-policy TD($\lambda$) panel, consider the other panels of Figure 4. Overall, there are a lot of similarities between the algorithms and how their performances varied with $\alpha$ and $\lambda$. For all algorithms, error was lower for $\lambda = 1$ (the blue line) than for $\lambda = 0$ (the red line). Bootstrapping apparently confers no advantage in the Collision task for any algorithm.

The most obvious difference between algorithms is that the performance of the two Emphatic-TD algorithms varied relatively little as a function of $\lambda$; their blue and red lines are almost on top of one another, whereas those of all the other algorithms are qualitatively different. The emphatic algorithms generally performed as well as or better than the other algorithms. At $\lambda = 1$, the emphatic algorithms reached the minimal error level of all algorithms ($\approx$0.1), and their ranges of good $\alpha$ values was as wide as that of the other algorithms. While at $\lambda = 0$, the best errors of the emphatic algorithms were qualitatively better than those of the other algorithms. The minimal $\lambda = 0$ error level of the emphatic algorithms was about 0.15, as compared to approximately 0.32 (shown as a second thin gray line) for all the other algorithms (except Proximal GTD2, a special case that we consider later). Moreover, for the emphatic algorithms the sweet spot for $\alpha$ shifted little as $\lambda$ varied. The shift was markedly less than for the six algorithms in the middle two rows of Figure 4. The lack of an interaction between the two parameter values is another potential advantage of the emphatic algorithms.

The lowest error level for eight of the algorithms was $\approx$0.1 (shown as a thin gray line), and for the other three algorithms the best error was higher, $\approx$0.16. The differences between the eight and the three were highly statistically significant, whereas the differences within the two groups were negligible. The three algorithms that performed worse than the others were Tree Backup($\lambda$), Vtrace($\lambda$), and ABTD($\zeta$)—shown in the bottom row of Figure 4. The difference was only for large $\lambda$s; at $\lambda = 0$ these three algorithms reached the same error level ($\approx$0.32) as the other non-emphatic algorithms. The three worse algorithms' range of good $\alpha$ values was also slightly smaller than for the other algorithms (with the partial exception, again, of Proximal GTD2($\lambda$)). A mild strength of the three is that the best $\alpha$ value shifted less as a function of $\lambda$ than for the other six non-emphatic algorithms. Generally, the performances of these three algorithms in Figure 4 look very similar as a function of parameters. An interesting difference is that for ABTD($\zeta$), we only see three gray curves, whereas for the other two algorithms we see seven. For ABTD($\zeta$) there is no $\lambda$ parameter, but the parameter $\zeta$ plays the same role. In our experiment, ABTD($\zeta$) performed identically for all $\zeta$ values greater than 0.5; four gray lines with different $\zeta$ values are hidden behind ABTD's blue curve.

In summary, our main result is that on the Collision task the performances of the eleven algorithms fell into three groups, or tiers. In the top tier are the two Emphatic-TD algorithms, which performed well and almost identically at all values of $\lambda$ and significantly better than the other algorithms at low $\lambda$. Although this difference did not affect best performance here (where $\lambda = 1$ is best), the ability to perform well with bootstrapping is expected to be important on other tasks. In the middle tier are Off-policy TD($\lambda$) and all the Gradient-TD algorithms including HTD($\lambda$), all of which performed well at $\lambda = 1$ but less well at $\lambda = 0$. Finally, in the bottom tier are Tree Backup($\lambda$), Vtrace($\lambda$), and ABTD($\lambda$), which performed very similarly and not as well as the other algorithms at their best parameter values. All of these differences are statistically significant, albeit specific to this one task. In Figure 4 the three tiers are the top row, the two middle rows, and the bottom row.

The reason why Emphatic TD algorithms reached a lower error level than some others might be the objective function they minimize. Emphatic TD algorithms minimize the Emphatic weighted Mean Squared Projected Bellman Error (MSPBE). This is in contrast to all other algorithms studied in this paper, that minimize the behavior policy weighted MSPBE. In our results, the error measure is the Mean Squared Value Error ($\overline{\text{VE}}$): the difference between the true value function and the value function found by an algorithm. Our results suggest that distance between the minimums of Emphatic weighted MSPBE and $\overline{\text{VE}}$ is smaller than the distance between the minimums of behavior weighted MSPBE and $\overline{\text{VE}}$.

The reason why ABTD, Tree Backup, and Vtrace did not perform as well as others is most probably that they cut off the importance sampling ratio. By cutting off the importance sampling ratio, these algorithms introduce bias into the solution that the algorithm finds, which in turn will cause the algorithm to converge to a higher error level. On the other hand, one can expect that these algorithms perform better than others on problems where importance sampling ratios are really large.

In the next two sections we take a closer look at two of the tiers to find differences within them.

## 8 Emphatic TD($\lambda$) vs. Emphatic TD($\lambda, \beta$)

In this section, the effect of the $\beta$ parameter of Emphatic TD($\lambda, \beta$) on the algorithm's performance in the full bootstrapping case is analyzed. We focus on the full bootstrapping case ($\lambda = 0$) because the largest differences were observed with this value of $\lambda$ in the previous section. The curves shown in Figure 4 in the previous section, are for the best values of $\beta$; meaning that, for each $\lambda$, we found the combination of $\alpha$ and $\beta$ that resulted in the minimum average error, fixed $\beta$, and plotted the sensitivity for that fixed $\beta$ over $\alpha$. Here, we show how varying $\beta$ affects performance.

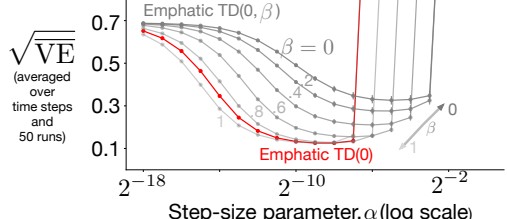

Figure 5: Detail on the performance of Emphatic TD($\lambda, \beta$) at $\lambda = 0$. Note that Emphatic TD($\lambda$) is equivalent to Emphatic TD($\lambda, \gamma$), and here $\gamma = 0.9$. The flexibility provided by $\beta$ does not help on the Collision task.

The error of Emphatic TD(0), and Emphatic TD(0,$\beta$) for various values of $\alpha$ and $\beta$ are shown in Figure 5. We see that both algorithms performed similarly well on the Collision task, meaning that they both had a wide sensitivity curve and reached the same ($\approx 0.1$) error level. Notice that, as $\beta$ increased, the sensitivity curve for Emphatic TD(0,$\beta$) shifted to left and the error overall decreased. With $\beta = 0$, Emphatic TD($\lambda, \beta$), reduces to TD($\lambda$). With $\beta = 0.8$, and $\beta = 1$, Emphatic TD($\lambda, \beta$) reached the same error level as Emphatic TD($\lambda$). With $\beta = \gamma$, Emphatic TD($\lambda, \beta$) reduces to Emphatic TD($\lambda$). This explains why the red curve is between the $\beta = 0.8$ and $\beta = 1$ curves.

The results make it clear that the superior performance of emphatic methods are almost entirely due to the basic idea of emphasis; the additional flexibility provided by $\beta$ of the Emphatic TD($\lambda, \beta$) was not important on the Collision problem.

## 9 Assessment of Gradient-TD Algorithms

We study how the $\eta$ parameter of Gradient-TD algorithms affects performance in the case of full bootstrapping (the second step size, $\alpha_{\mathbf{v}}$, is equal to $\eta \times \alpha$). Previously, in Figure 4 we looked at the results with the best values of $\eta$ for each $\lambda$; meaning that for each $\lambda$, first the combination of $\alpha$ and $\eta$ that resulted in the lowest average $\overline{\text{VE}}$ was found and then sensitivity to step size was plotted for that specific value of $\eta$. Sensitivity to step size for various values of $\eta$ with $\lambda = 0$ are shown in Figure 6. Each panel shows the result of two Gradient-TD algorithms for various $\eta$s: One main algorithm, shown with solid lines, and another additional algorithm shown with dashed lines for comparison. First focus on the upper left panel. The upper left panel shows the parameter sensitivity for GTD2(0), for four values of $\eta$, and additionally it shows GTD(0) results as dashed lines for comparison (for results with more values of $\eta$ see Appendix D). The color for each value of

$\eta$ is consistent within and across the four panels, meaning that for example, $\eta = 256$ is shown in green in all panels, either as dashed or solid lines. For all parameter combinations, GTD errors were lower than (or similar to) GTD2 errors. With two smaller values of $\eta$ (1 and 0.0625) GTD had a wider and lower sensitivity curve than GTD2, which means GTD was easier to use than GTD2.

Let us now move on to the upper right panel of Figure 6. Proximal GTD2 had the most distinctive behavior among all Gradient-TD algorithms. As previously observed in Figure 3, it is the only algorithm that in some cases had a "bounce"; its error dipped down at first and then moved back up. With $\lambda = 0$, in some cases it converged to a lower error than all other Gradient-TD algorithms. Proximal GTD2 was more sensitive to the choice of $\alpha$ than other Gradient-TD algorithms except GTD2. Proximal GTD2 had a lower error and a wider sensitivity curve than GTD2. To see this, compare the dotted and solid lines in the upper right panel of Figure 6.

Moving on to the lower left panel, we see that GTD and HTD performed similarly. Sensitivity curves were similarly wide but HTD reached a lower error in some cases. We see this by comparing the dotted and solid pink curves in the lower left panel.

The fourth panel shows sensitivity to the step-size parameter for HTD and TDRC. Notice that TDRC has one sensitivity curve, shown in dashed blue. This is because $\eta$ is set to one (also its regularization parameter was set to one) as proposed in the original paper. HTD's widest curve was with $\eta = 0.0625$ which was as wide as TDRC's curve. For a more in-depth study of TDRC's extra parameters see Appendix D.1.

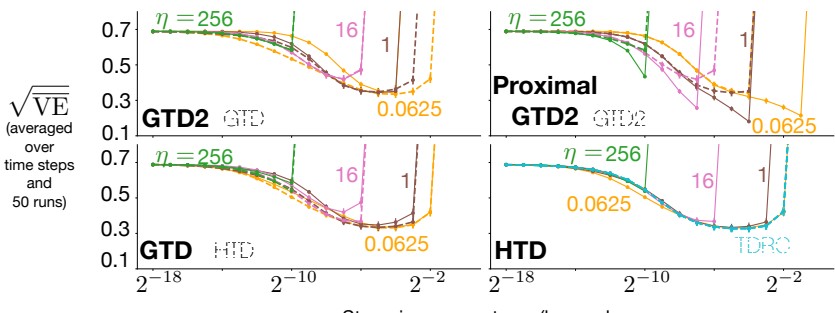

Figure 6: Detail on the performance of Gradient-TD algorithms at $\lambda = 0$. Each algorithm has a second step-size parameter, scaled by $\eta$. A second algorithm's performance is also shown in each panel, with dashed lines, for comparison.

On one hand, among the Gradient-TD algorithms, TDRC was the easiest to use. On the other hand, in the case of full bootstrapping, Proximal GTD2 reached the lowest error level. The fact that proximal GTD2 converged to a lower error level might be due to a few different reasons. One possible reason is that it might not have converged to the minimum of the mean squared projected Bellman error like other Gradient-TD algorithms. Another reason might be that it converged to a minimum of the projected Bellman error that was different from the minimum the other algorithms converged to. Further analyses is required to investigate this. It remains to be seen how these algorithms compare on other problems.

## 10    Limitations and Future Work

The present study is based on a single task, and this limits the conclusions that can be fairly drawn from it. For example, we have found that Emphatic-TD methods perform well over a wider range of parameters than Gradient-TD methods on the Collision task, but it is entirely possible that the reverse would be true on a different task. Many more tasks must be explored before it is possible for a consistent pattern to emerge that favors one class of algorithm over another.

On the other hand, a pattern over empirical results must begin somewhere. We stress the need for extensive empirical results even for a single task. Ours is the first systematic study of off-policy learning to describe the effects of all algorithm parameters individually (rather than, for example, taking the best performing parameters or fixing one parameter and studying another). Such a thorough examination is necessary to obtain the understanding that is critical to using off-policy algorithms successfully and with confidence. There is a need for thorough empirical studies, but they take time, and a proper presentation of them takes space.

While our study is not the last word, it does contribute to the growing database of reliable results comparing modern off-policy learning algorithms.

Conducting additional experiments with other off-policy learning problems is a valuable direction for future work. In looking for the next problem, one might seek a task with greater challenges due to variance of the importance sampling ratios. In the Collision task, the product of ratios can grow nearly as large as $2^4 = 16$. This could be made more extreme simply by increasing the number of states, or by changing the behavior policy. Also valuable would be exploring unrelated tasks with a different rationale for relevance to the real world. One possibility is to use a task related to parallel learning about multiple alternative ways of behaving, such as learning how to exit each room in a four-rooms gridworld (Sutton, Precup & Singh, 1999).

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
