# OpenReview forum: "An Empirical Comparison of Off-policy Prediction Learning Algorithms on the Collision Task"
_TMLR — Rejected by TMLR_

### Review · Reviewer_dyR6 · 2022-07-27

**Summary Of Contributions:**

This paper focused on the empirical study of learning predictions in reinforcement learning (RL) algorithms, particularly in the multi-step off-policy setting. To do that, eleven off-policy learning algorithms are selected, including the algorithm families of gradient-TD, emphatic-TD, off-policy TD, Vtrace, and variants of Tree Backup and ABQ, with experiments being conducted on a single prediction problem of "collision task". Finally, their performance was accessed through the mean square value error, especially from the perspectives of the learning rate, asymptotic error level, sensitivity to step size, and bootstrapping parameters.


**Broader Impact Concerns:**

This paper focused on the simulated experimental environment. So I don't have any concerns regarding this.


**Requested Changes:**

- I have major concerns about the motivation and contribution of this paper as mentioned in the main weaknesses. Overall, I don't think the current status of the paper is ready.  I suggest the authors address them first.
- I also have some additional comments and the authors could check them and update the missing reference.


**Strengths And Weaknesses:**

Strengths:
- Learning the value function is critical to RL. This paper targets learning predictions for one policy while following another policy with eligibility traces, which will be interesting to the RL community.

Weaknesses:
- This work is not well-motivated. It's not clear why those existing eleven linear-complexity algorithms are selected to make a comparison without proposing any new approaches. What's more important, it's not clear what specific problems in the existing eleven algorithms or what properties those algorithms have, or even what scenarios those algorithms can fit in.
- The contributions look quite limited. There is no new algorithm proposed in this paper, although it claims in this paper that it treats more algorithms and does a deeper empirical analysis, compared to the previous work of White and White (2016). Actually, the current empirical analysis is limited to comparing the performance on a simple, small domain (collision task) in terms of the learning rate, error level, sensitivity to step size, and bootstrapping parameters, which is not convincing and deep enough. It should at least evaluate those algorithms on more different tasks with different complexity and show their performance regarding convergence/ standard deviation among multiple runs, etc.


Some additional comments:
- Regarding the collision task, since the target policy is to keep forwarding from state 5 to state 8 while the corresponding behavior policy is uniform, how this task can help exit a parallel parking spot without collision, or learn to get close to other cars without collisions? Furthermore, how to treat the previous two examples of avoiding collisions as off-policy learning problems and learn with fewer collisions?
- All algorithms of Tree Backup, Retrace, and ABQ are evaluated on the policy evaluation tasks in their original papers, why does this paper need to derive their prediction variants? Are there any reasons for doing it in a different way?
- The reference to "Chung et al. (2018)" is missing.
- Why not select and compare those eleven algorithms with GTB and GRetrace as shown in Touati et al. (2018)?
- Is this $\overline{\rm VE}(0) \approx 0.7$ correct? According to the description, should it be the square root of $\overline{\rm VE}(0)$ to be $0.7$?
- In Figure 1, what is the algorithm running for this example?

---

> ### Author Response · Authors · 2022-07-28
> **Response to Reviewer dyR6 concerns**
>
> Thanks for the review. The reviewer has two main concerns: motivation and scope of contributions. We address each below.
>
> # Motivation:
>
> The motivation of the work is simple: many off-policy learning algorithms have been proposed, but it’s not clear how they compare to each other. We are not the first to focus on assessing the performance of existing algorithms and our study is not the last word, but it does contribute to the growing database of reliable results comparing modern off-policy learning methods.
>
> # Limitation of contribution:
>
> We agree that significance and limitation of scope are at times subjective. The reviewer is concerned that:
> 1) no new algorithm is proposed in the paper,
> 2) the empirical analysis is limited to comparing the performance on a small domain in terms of learning rate, asymptotic error, and sensitivity to parameters, and
> 3) the algorithms should be compared on more domains, and average performance over multiple runs should be measured.
>
> We begin by reiterating our main contribution: insights gained from a detailed empirical study. We don’t believe for a paper to make a contribution, a new algorithm is a necessity. A study of existing methods helps the community understand algorithms’ differences and interrelationships and helps direct future developments. To determine if a paper makes a contribution, the main question is if it changes our understanding or not.
>
> We’d like to invite the reviewer to elaborate on why an empirical study of methods’  learning rate, error level, and sensitivity to parameters is not “convincing and deep enough”. As discussed in the paper, we agree that conducting more experiments is a valuable direction for future research. However, this does not mean the current work does not add to what the field knows about how existing algorithms perform. Regarding the small size of the problem, there is a trade-off between problem size, and how detailed the experiments can be. We chose a small size problem to be able to go into significant detail.
>
> We'd like to ask the reviewer to go over the last section of the paper once more:
> > The present study is based on a single task, and this limits the conclusions that can be fairly drawn from it. Many more tasks must be explored before it is possible for a consistent pattern to emerge that favors one class of algorithm over another. On the other hand, a pattern over empirical results must begin somewhere. Ours is the first systematic study of off-policy learning to describe the effects of all algorithm parameters individually, rather than, for example, taking the best performing parameters or fixing one parameter and studying another. There is a need for thorough empirical studies, but they take time, and a proper presentation of them takes space.
>
> Regarding comparing algorithms over multiple runs, we reiterate that all results reported in the paper are averaged over 50 trials, with standard errors on all curves.
>
> Finally, we’d like to end by reiterating the two main TMLR’s acceptance criteria:
> 1) will the paper be interesting to some readers, and
> 2) are the main claims of the paper well-supported?
>
> Fortunately, the reviewer has determined that the paper will be of interest to the RL community. The reviewer also did not raise any concerns about any of the main claims of the paper. This hopefully makes the paper a good candidate for publication at TMLR.
> # Minor concerns:
> * About learning through less collisions:
>   - The goal is not to learn without collisions, but through fewer collisions. We will sharpen up this point in the final version.
> * Tree Backup, Retrace, and ABQ are evaluated on the policy evaluation tasks in their original papers, why does this paper need to derive their prediction variants?
>   - This is not true. Retrace, ABQ, and Tree Backup are all control algorithms for learning state action pairs and not state values. This is partly why the Vtrace algorithm (prediction variant of Retrace) is developed in the IMPALA paper (Espeholt 2018). Subtle changes are necessary before ABQ and Tree Backup can be applied to learn state values. The prediction variants of Tree Backup and ABQ are the smaller contributions of our paper.
> * Why not include GTB and GRetrace?
>   - Many algorithms mentioned in our paper can be combined with each other; e.g, combination of ETD and GTD makes Emphatic Gradient TD. Similarly, GTD+Tree Backup makes GTB (Touati, 2018–same for GRetrace). We didn’t include algorithm combinations to limit the paper’s scope. Having said that, we will be happy to include GTB and GRetrace in the final version.
> * What’s the algorithm in Fig.1?
>   - Algorithm names are intentionally omitted from Fig.1. The purpose of Fig.1 is not comparing algorithms, but to the range of possibilities on the task. To know the range of possibilities, the name of the algorithm is not necessary and might in fact be confusing. Fig.1’s caption reads: “Learning curves illustrating the range of things that can happen during a run”.

---

> > ### Author Response · Authors · 2022-08-04
> > **Response to Reviewer dyR6 (part 2)**
> >
> > We improved the paper by addressing some of the issues that reviewer dyR6 raised. Specifically:
> > - We added the missing reference to Chung et al., 2018 (see the green reference added to the References section).
> > - We added a point about learning through fewer collisions. _Please see the green and red text we added to Section 5_.
> > - We added a few sentences to address why the algorithms of Touati et al. (2018) are not included. _Please see the green text in Section 3 of the paper_.

---

> > ### Comment · Reviewer_dyR6 · 2022-08-20
> > **Posts after response**
> >
> > I appreciate the long response from the authors. I carefully read the response and the revised version of the manuscript as well. However, most of my concerns remain unaddressed:
> > - motivation: This paper aims to achieve some reliable results for existing off-policy learning algorithms. However, it's still unclear why those multi-step off-policy prediction algorithms are selected. Why the previous results of those algorithms in their original papers were not reliable? Or are there any specific purposes or scenarios that those algorithms can fit in?
> > - limited contribution: It's not true that only brand new algorithms can be treated as a contribution. But it is indeed necessary to clarify what are the differences between this paper and the previous work of White and White (2016), and why the differences can be significant. In its current form of empirical analysis, the test domain is not representative enough and I am wondering if the current conclusion can be generalized to other different domains. For the depth of empirical analysis, here are some examples [1,2] that can clearly identify their contributions, which are empirical analysis papers as well.
> > - prediction variants: I am sure that Tree Backup, Retrace, and ABQ were evaluated on the policy evaluation tasks in their original papers. It's still unclear why this work needs their extra prediction variants.
> >
> >
> > References:
> > - [1] Dann, C., Neumann, G., & Peters, J. (2014). Policy evaluation with temporal differences: A survey and comparison. Journal of Machine Learning Research, 15, 809-883.
> > - [2] Voloshin, C., Le, H. M., & Yue, Y. (2019). Empirical analysis of off-policy policy evaluation for reinforcement learning. In Real-world Sequential Decision Making Workshop at ICML (Vol. 2019).

---

> > > ### Author Response · Authors · 2022-08-20
> > > **Response to reviewer's concerns**
> > >
> > > Thanks for your response. We address each concern below, in the same order the concerns were brought up.
> > >
> > > ### **_Motivation:_**
> > > **_Regarding the set of chosen algorithms:_** We tried to choose as many algorithms as possible that fit within our criteria: fully incremental online off-policy prediction learning. Having said that, we are absolutely open to adding any new algorithms that the reviewer might believe is necessary to make the results of the paper comprehensive.
> > >
> > > **_Regarding the motivation and why results to date are not sufficient:_** Our study is an in-depth _comparative_ study. We actually believe that the results provided in previous studies are in fact reliable, but, we believe, as we mentioned in the paper: “ We are not the first to focus on assessing the performance of existing algorithms and our study is not the last word, but it does contribute to the growing database of reliable results comparing modern off-policy learning methods”. The reviewer probably agrees with us that the field does not yet know in what situation, which off-policy learning algorithm should be preferred. If this is agreed upon, it is clear that more reliable results are required before the relative merits of these algorithms are understood.
> > >
> > >
> > > ### **_Limitation:_**
> > > As we mentioned in the last paragraph of the introduction section of the paper, the main difference between our work and White and White (2016) is that: “Our study differs from earlier studies in that it treats more algorithms and does a deeper empirical analysis on a single problem, the Collision task. The additional algorithms are the prediction variants of Tree Backup (Precup, Sutton, & Singh, 2000), Retrace (Munos et al., 2016), ABQ (Mahmood, Yu, & Sutton, 2017), and TDRC (Ghiassian et al., 2020). Our empirical analysis is deeper primarily in that we examine and report the dependency of all eleven algorithms’ performance on all of their parameters individually [as opposed to other studies that maximize over one of the parameters]. This level of detail is needed to expose our main result, an overall ordering of the performance of off-policy algorithms on the Collision task. Our results, though limited to this task, are a significant addition to what is known about the comparative performance of off-policy learning algorithms”.
> > >
> > > As the reviewer mentioned, the results in this study will _not_ directly transfer to _all_ other tasks, and this itself is an excellent reason why more reliable empirical results are necessary.
> > >
> > >
> > > ### **_Prediction variants of ABQ, Tree Backup, and Retrace:_**
> > > With ABQ, Tree Backup, and Retrace, it is not possible to learn state-values in an off-policy manner. However, it is possible to learn state-action pair values. We derived the prediction variant of these algorithms to be able to directly learn state-values, like all other algorithms in our study. This is similar to what the V-trace algorithm in the IMPALA paper (Espeholt et al., 2018) does. V-trace is the variant of the Retrace algorithm capable of learning state-values instead of state-action pair values. We will make sure this point is clear in the final version of the manuscript.
> > >
> > > The points that we would like to stress at the end are:
> > > - As the reviewer mentioned in their first review, our results will be of interest to the RL community, and;
> > > - no concerns are brought up regarding the soundness of the results.
> > >
> > > We hope that soundness of the results and the interest of the RL community, makes our paper a good candidate for publication at TMLR.
> > >
> > > _References:_
> > > - Espeholt, et al. "Impala: Scalable distributed deep-rl with importance weighted actor-learner architectures." International conference on machine learning. PMLR, 2018.

---

### Review · Reviewer_owx3 · 2022-07-31

**Summary Of Contributions:**

The submission performs an empirical analysis of *off-policy prediction* of 11 off-policy RL algorithms on a simple RL task called *collision*. Off-policy prediction consists is learning the value function for a *target* policy with data collected with another *behavioral* policy. To the best of my understanding, this is what is more commonly called off-policy policy evaluation. Collision is a finite MDP with 8 states and two actions. The RL algorithms use a linear state representation of dimension 6, thus preventing a complete independence between the states. The 11 algorithms are $TD(\lambda)$, $GTD(\lambda)$, $GTD2(\lambda)$, Proximal $GTD2(\lambda)$, $HTD(\lambda)$, $TDRC(\lambda)$, $Emphatic$ $TD(\lambda)$, $Emphatic$ $TD(\lambda,\beta)$, $ABTD(\zeta)$, $Vtrace(\lambda)$, and $Tree$ $Backup(\lambda)$.

**Requested Changes:**

Please address my concerns enumerated in the weaknesses.

**Strengths And Weaknesses:**

--- Strengths ---
* The writing is methodological, mostly typo-free.
* The paper is overall well presented.

--- Weaknesses ---

**Major importance:**
* I would like to understand the difference of the task with that of off-policy policy evaluation. There has been plenty of papers and dedicated algorithms on this topic, which are mostly ignored (https://web.mit.edu/6.246/www/lectures/Off-policy.pdf). The off-policy algorithms used in the benchmark are off-policy RL algorithms, meaning that they intend to optimize the target policy at the same time they learn their value function. Why evaluate these algorithms on the off-policy value prediction task, which is merely an auxiliary task for them? Also there has been an empirical benchmark on off-policy policy evaluation: https://openreview.net/pdf?id=IsK8iKbL-I . It would be useful to detail your contribution with respect to theirs.
* The empirical study is very narrow: a single environment with 8 states and 2 actions, no stochasticity. I notice that the authors try to defuse this criticism, and I would agree with them in general: it is generally better to have clear insights on a constrained setting than vague conclusions on a broad task. But here, the analysis of the why is missing. We observe clear trends which are extensively described, but the paper does not provide any clue why some algorithms fail while others work. As a consequence, the contribution is limited to the very domain of collision.
* Some bits of the exposition are missing or imprecise or inaccurate:
  * it is unclear during the whole introduction whether the algorithms are compared on their actual purpose (off-policy RL) or the off-policy prediction task.
  * the importance sampling ratio is not introduced / formally defined.
  * Page 5, in the definition of $G^\lambda_t$, $\delta_t$ should be $\delta_n$, and the notation used is very confusing as it can be interpreted as algebra and useless as it's longer than the full writing of the product.
  * why $\rho_t \lambda_{t+1}$ and not $\rho_t \lambda_t$? Why is it well-behaved? I am not very knowledgeable in linear off-policy TD-learning algorithms, and the exposition did not help me to construct a better map of them, because the specificities of each algorithm is not motivated.

---

> ### Author Response · Authors · 2022-08-04
> **Response to Reviewer owx3 concerns**
>
> Thanks for the detailed and constructive review. The reviewer brought up three major points. We respond to the points in the same order they were brought up.
>
> - The first concern is a misunderstanding and will be addressed by a small change in presentation. Off-policy evaluation (OPE) can in general be done online or offline. In the literature, however, the word OPE is often used when learning is _offline_ and _off-policy_. In the offline learning setting, the agent learns with access to data beyond what it experiences at each time step. In this setting, the agent has either access to historical data, or a body of data gathered through interaction with the environment. All algorithms that are based on doubly robust estimators are examples of this setting. In contrast, we studied the setting that is sometimes referred to as the online fully incremental RL. Examples of papers that only focus on fully incremental methods are the ones that we discussed in the introduction, in the last paragraph of page 2. In online fully incremental learning, the assumption is that the agent receives a sample, learns from it, and then disposes of the sample. This is what Sutton and Barto (2018) focus on in most of their RL book. We agree with the reviewer that it is valuable to study other off-policy learning algorithms and other settings, like doubly robust algorithms, but we chose to focus on fully incremental online algorithms to focus the scope of the paper, in order to be able to go into great detail. _Please see the blue text that we added to Section 1 of the paper to make this point clear_.
>
> - The second concern of the reviewer is that the study is narrow. We agree that our study is focused, but at the same time, we believe that it is only possible to go into the amount of detail that we did, given that the study is focused. It is valid that the reviewer is interested in knowing **why** the algorithms perform the way they did. We agree that adding some information on the why part will make the paper stronger and thus we added some discussion in the paper on why some of the results turned out the way they did. _Please see the blue text that we added to Section 7 of the paper_. Unfortunately, we don’t think it’s possible to investigate why all the 11 algorithms performed the way they did in a single paper as this includes solving many open problems. For example, it is not at all clear why ETD converges to a lower error level than others. We know that ETD converges to the minimum of the emphatic weighted MSPBE, while our error measure is MSVE. GTD algorithms converge to the minimum of behavior policy weighted MSPBE. To show why ETD converges to a lower error than GTD, would be to show why the minimum of Emphatic weighted MSPBE is closer to the minimum of MSVE, compared to the minimum of the behavior weighted MSPBE. This is a very interesting question, whose answer most probably needs a paper of its own.
>
> - Minor comments:
>    * Thanks for pointing out that the importance sampling ratio is not included. We corrected this. _Please see the blue text after equation 4 in Section 4_.
>    * We added a subsection to the paper that hopefully makes section 5 more clear and easy to understand. It includes the reasoning why we used $\rho_t\lambda_{t+1}$ and not $\rho_t\lambda_t$. _Please see the blue text that we added to Section 4.1 of the paper to make this point clear_.

---

### Review · Reviewer_MNQm · 2022-08-03

**Summary Of Contributions:**

The paper provides an empirical evaluation of eleven off-policy prediction algorithms on a specific problem, the collision task. Since some of the approaches to be tested are designed for control, the authors first rephrase them into their prediction version, devising suitable update rules. Then, the collision task, a tabular domain, is presented and described in its features. Finally, the experimental evaluation is performed; each combination (algorithm, hyperparameters) is tested over 50 runs and the performance is evaluated in terms of value error. Specifically, the first set of experiments aims at comparing all the algorithms by varying the step-size $\alpha$ and the $\lambda$ parameter related to the $\lambda$-return. Successive experiments aim to dive into the peculiarities of emphatic TD methods and gradient TD algorithms. A final discussion on limitations and future works is present.

**Broader Impact Concerns:**

None.

**Requested Changes:**

**Critical Requested Changes**

* The authors should provide convincing motivations for their choices. I am referring, in particular, to my comments (Tabular Task and Value Function Approximation), (Construction of the Random Features), (Weight Initialization). As an alternative, if no motivation can be provided, the authors could empirically evaluate (if this is feasible in terms of computational time requested) multiple alternatives for these choices (e.g., testing the exact tabular representation, re-run the experiments with randomly generated weights).

* The authors should clarify the concern (About the Collision Task) regarding the reward function.

**Non-Critical Requested Changes**

* Fix minor issues.

**Strengths And Weaknesses:**

**Strengths**

* The paper provides, to the best of my knowledge, the first empirical comparison that includes a significantly large number of baselines (11), including some that are designed for control and rephrased for prediction.

* The experimental evaluation is well conducted from a methodological perspective. Specifically:
- Each (algorithm, hyperparameters) is tested multiple times (50 runs) and the total number of configurations is notably large (20k).
- The plots include appropriate error bars, although (from the main paper) it is not clear what they represent. Standard deviation, confidence intervals?

 *The supplementary material includes the code and a ReadMe that describes the implementation of the algorithms, the implementation of the environment, and the instructions to run the code (note that I did not re-run the code). Moreover, the full range of hyperparameters tested is reported.

* The authors demonstrate that they are aware of some of the limitations of the present work, especially the fact the experimental evaluation is performed in one environment only.

**Weaknesses**

* (About the Collision Task) If I understood well, the collision task aims at modeling the scenario in which a vehicle moves towards an obstacle and has to avoid it. If this is correct, I found the selected reward function quite unexpected. Indeed, if the vehicle decides to turnaway avoiding the obstacle, the episode terminates and the reward is $0$. If instead, the vehicle decides to hit the obstacle, the episode also terminates, but the reward is $+1$. Isn't it convenient to hit the obstacle with such a reward function? Can the authors clarify? Moreover, the description of the environment in the caption of Figure 1 is not very accurate (especially regarding the reward function).

* (Tabular Task and Value Function Approximation) The considered task is a tabular one, made of 8 states and 2 actions (not available in all states). Nevertheless, the authors decided to use linear value function approximation with randomly generated features (generated with a specific criterion, see comment below). This seems quite artificial, as also the authors acknowledge. The authors motivate their choice by claiming that a tabular approach is not feasible in large problems. While I agree with the claim, the Collison task is not a large problem, and I think the statement does not justify the choice. It would have been more convincing to start from a continuous-state problem directly. Is there a specific reason why to use a tabular problem for this experimental evaluation?

* (Construction of the Random Features) The authors model the fact that in real-world scenarios, the agent accesses sensors whose measurements then pass through a neural network, by considering a linear value function approximator with randomly generated features. Those features are 6-dimensional, binary, and randomly generated. In such a way, since the problem has 8 states, it is not possible to get a non-ambiguous representation. I appreciate the effort in coming up with an approach to model a complex phenomenon (sensor measurements + NN), but the proposed choice seem quite arbitrary. Why 6 features? Why binary? In my view, the authors should provide a convincing motivation behind these choices or test multiple alternatives (including the tabular representation).

* (Weight Initialization) The authors decided to initialize the weights to zero. Is there a specific reason behind this choice? If the overall goal is to simulate what happens when training with NNs, shouldn't the weights be initialized randomly?

**Summary**
Although the paper has elements of appreciable novelty (as noted above), I identified several limitations that I will re-list in the following for convenience:
* Evaluation conducted on just one task (acknowledged by the authors too);
* Tabular task addressed with value function approximation;
* Not-convincing construction of the features for the linear value function approximation;




**Minor Issues**
* Section 1 presents both the related works and the contributions. I suggest either moving the related works to another section or, at least, denoting with suitable subsections/titles the part of related works and that of contributions.

* When multiple citations are listed in a row, I suggest sorting them by year.

* Some full-line equations miss the punctuation (e.g., the equation right before equation (1).

---

> ### Author Response · Authors · 2022-08-04
> **Response to reviewer MNQm concerns**
>
> Thank you for the detailed and constructive feedback. We respond to each point you brought up separately:
>
> - **_Why construct random features?_**
> As shown in the literature, the effect of feature representation on learning is extremely high. Although it was impossible for us to study the effect of all different kinds of feature representations on the learning process, we chose to focus on one of the major feature representation schemes (one hot representation–see chapter 9 of Sutton and Barto’s RL book). We chose one-hot features, because we wanted to have state-aliasing, meaning that some features are shared between states. We chose 6 features because we wanted it to be impossible for the states to be exactly represented in the feature space. At the end, we could choose 5, or some other number, but it was important for us to have state-aliasing, and approximation. Focusing on one-hot representations, we used a different random representation for each of the 50 runs, to study the performance of algorithms across various one-hot representations. We believe that more than one study is required to understand the effects of feature representation on learning, even on a small problem like ours. _Please see the red text added to Section 6_.
>
> - **_Why initialize weights to all zeros?_**
> In deep learning, the weights are not all initialized to zeros, because if they are, the derivatives in backprop will be the same for all weights, which in turn means that all the learned features will be the same. Such a problem does not exist in our case, because we are using fixed features (in a linear function approximation setting). Having said that, we’ll be happy to add random initialization results (during possible revision period) to the appendix of the paper if the reviewer sees this as necessary. _Please see the red text added to Section 6_.
>
> - **_Why choose such a reward function? Will the agent learn to hit the wall more with this reward function?_**
> Please note that our task is a prediction, and not a control task. If the task was a control task, the car would learn to hit the obstacle more often, but in our setting, the behavior and target policies are fixed and given, and the goal is to only learn about collisions. These predictions can later be used for different purposes such as state construction, and control. We agree with the reviewer that it would’ve been more intuitive if the reward was -1 when the car hits the obstacle; however, we decided to keep the reward function as is because the goal of this experiment is for the agent to learn about hitting an obstacle, regardless of the reward being positive or negative. _Please see the red text added to Section 5_.
>
> - **_Why evaluate on one task only?_**
> We simply chose to focus on one task, to be able to go into appropriate detail necessary to make a meaningful comparison of algorithms over all their parameters. We also tried to answer this question in the conclusion of the paper, as we believed others might have this question as well: “The present study is based on a single task, and this limits the conclusions that can be fairly drawn from it. Many more tasks must be explored before it is possible for a consistent pattern to emerge that favors one class of algorithm over another. On the other hand, a pattern over empirical results must begin somewhere. Ours is the first systematic study of off-policy learning to describe the effects of all algorithm parameters individually (rather than, for example, taking the best performing parameters or fixing one parameter and studying another). There is a need for thorough empirical studies, but they take time, and a proper presentation of them takes space”.
>
> - **_Motivation for tabular task with linear function approximation (FA)_**
> We looked at a small task to be able to go into great detail. This means a small number of states and actions; which in turn means suitable for tabular settings. However, we did not want all algorithms to be able to solve the task perfectly, which would be the case in tabular settings. With tabular features, all algorithms would be able to solve the task perfectly and reach zero error. For example, in our experiment, ETD converged to a lower error than GTD. ETD minimizes the Emphatic weighted MSPBE, and GTD minimizes the behavior weighted MSPBE. These two objective functions only differ if FA is used. In the tabular case, both objectives have a 0 minimum. Many differences observed in the paper are contingent on FA. The purpose of the paper was not to solve the task in the most convenient way, but to design a task that is capable of bringing out algorithms’ differences. Regardless, we will be able to add tabular results to the appendix of the paper if the reviewer finds it important.
>
> - **_What are the error bars on figures?_** As mentioned on the 8th line of the second paragraph of section 8, the bars represent standard error. _For better clarity, we added this to Figure 4 caption in red_.

---

### Author Response · Authors · 2022-08-15
**Have all the concerns been addressed?**

We thank all our reviewers for their time and constructive feedback. We have uploaded a revised version of the manuscript containing some clarifications and additions, as suggested by the reviewers.

As the discussion period is coming to an end, we'd like to invite the reviewers to have a second look at the updated manuscript and our responses to ensure all their concerns have been addressed. Please do reach out should there be any remaining questions.

---

### Decision · Action_Editors · 2022-09-18

**Recommendation:** Reject

**Comment:**

The paper provides an empirical analysis of several off-policy prediction algorithms, using as a benchmark a simple collision task.
Although the paper contains some interesting contributions, the reviewers have expressed several concerns about the lack of motivation for several choices and using a single environment for the evaluation, which is not enough to reach convincing conclusions, thus making the findings of the paper not of much interest for the TMLR audience.
The authors' rebuttals and the new version of the paper have solved some of the reviewers' concerns, but the reviewers agree that the paper is not ready for publication.
I encourage the authors to generate a new, significantly improved version of their paper that addresses the main issues highlighted by the reviewers and resubmit it.